# EEG Classification of Motor Imagery Using a Novel Deep Learning Framework

**DOI:** 10.3390/s19030551

**Published:** 2019-01-29

**Authors:** Mengxi Dai, Dezhi Zheng, Rui Na, Shuai Wang, Shuailei Zhang

**Affiliations:** 1School of Instrumentation and Optoelectronic Engineering, Beihang University, Beijing 100191, China; daimengxi@buaa.edu.cn (M.D.); RUI_NA@buaa.edu.cn (R.N.); zhangshuailei@buaa.edu.cn (S.Z.); 2Beijing Advanced Innovation Center for Big Data-based Precision Medicine, Beihang University, Beijing 100191, China; 3School of Computer Science and Engineering, Beihang University, Beijing 100191, China

**Keywords:** EEG, deep learning, short-time Fourier transform, convolutional neural network, variational autoencoder

## Abstract

Successful applications of brain-computer interface (BCI) approaches to motor imagery (MI) are still limited. In this paper, we propose a classification framework for MI electroencephalogram (EEG) signals that combines a convolutional neural network (CNN) architecture with a variational autoencoder (VAE) for classification. The decoder of the VAE generates a Gaussian distribution, so it can be used to fit the Gaussian distribution of EEG signals. A new representation of input was developed by combining the time, frequency, and channel information from the EEG signal, and the CNN-VAE method was designed and optimized accordingly for this form of input. In this network, the classification of the extracted CNN features is performed via the deep network VAE. Our framework, with an average kappa value of 0.564, outperforms the best classification method in the literature for BCI Competition IV dataset 2b with a 3% improvement. Furthermore, using our own dataset, the CNN-VAE framework also yields the best performance for both three-electrode and five-electrode EEGs and achieves the best average kappa values 0.568 and 0.603, respectively. Our results show that the proposed CNN-VAE method raises performance to the current state of the art.

## 1. Introduction

The brain-computer interface (BCI) is an alternative method of communication between a user and system that depends on neither the brain’s normal output nerve pathways nor the muscles. The process generally begins by recording the user’s brain activities and continues to signal-processing to detect the user’s intentions. Then, the appropriate signal is sent to the external device, which is then controlled according to the detected signal. One of the important goals of BCI research is to enhance certain functions for a healthy person via a new signal pathway. BCI is currently being studied in a fairly wide application range, including communication tools for locked-in state (CLIS) patients [1,2,3].

Studies have proved that the electroencephalogram (EEG) signals produced while mentally imagining different movements can be translated into different commands. In this paper, we refer to the brain potentials related to motor imagery tasks as control signals. In motor imagery (MI)-based BCI systems, the imagining of body movements is accompanied by a circumscribed event-related synchronization/desynchronization (event-related synchronization (ERS)/event-related desynchronization (ERD)) [4,5]. Thus, motor imagery has had widespread use as a major approach in BCI systems [6,7].

Pattern recognition techniques enable information extraction from EEG recordings of brain activity, which is why it is extremely important in EEG-based research and applications. Machine learning is the core of EEG-based BCI systems. Many studies have investigated numerous feature extraction and classification approaches to MI task recognition. In MI-based BCI systems, common spatial pattern (CSP) is a very popular and powerful feature extraction method [8,9]. It constructs very few new time series whose variances contain the most discriminative information. In recent years, the CSP algorithm has been further developed. Several approaches have been proposed, including filter bank CSP (FBCSP) [10], augmented common spatial pattern (ACSP) [11], etc. For the classification component, several major model types, such support vector machine (SVM) [12], linear discriminant analysis (LDA) [13] and Bayesian classifier [14] have been used [15].

Nevertheless, despite all the improvements that have been achieved in BCI systems, there is still room for enhancement, especially in interpretability, accuracy, and online application usability. To address these issues and construct a pattern recognition model with high descriptive power, a new category of strategies and methods, called deep learning, was recently developed, and it is prevalent in both academia and industry [16,17].

Deep learning methods have been rarely applied to the EEG-based BCI system, as they are quite hard to apply to the development of a perfect EEG classification framework due to various impacting factors, such as noise, the correlation between channels, and the high dimensional EEG data. An ideal classification framework should contain a data preparation stage, where signals undergo a reduction in dimensionality and are represented by new data without significant information loss. Subsequent to data preparation is the second stage of this framework-that is, the network architecture-where meaningful features of the inputs are all extracted. Taking into account all the challenges mentioned above, the successful implementation of deep learning methods for the classification of EEG signals is quite an achievement.

In one study [18], a Boltzmann Machine model was applied for two-class MI classification. Convolutional neural networks (CNN) are also used in some EEG studies. For instance, in [19], three CNNs with different architectures were used to classify MI EEG signals, with the number of convolutional layers ranging from 2 layers to a 5-layer deep ConvNet up to a 31-layer residual network (ResNet). In a recent study [20], a network was proposed which employs a recurrent convolutional neural network (RCNN) architecture to understand the cognitive events in EEG data. A new form that combines CNN and stacked autoencoders (SAEs) was proposed in [21] for MI classification.

Several studies have used different methods to convert EEG signals to an image representation before applying a CNN. In [10], filter bank common spatial pattern (FBCSP) features were generated based on pairwise projection matrices. This can be considered an extraction of the CSP features from a multilevel decomposition of different frequency ranges. In [20], the authors proposed a new form of features that preserves the spatial, spectral, and temporal structure of the EEG. In this method, the power spectrum of the signal from each electrode is estimated and the sum of squared absolute values are computed for three selected frequency bands. Then, the polar projection method is used to map the electrode locations from 3D to 2D and construct the input images.

A new form of input that combines frequency, time, and multichannel information is introduced in this study. The time series of an EEG is converted into two-dimensional images with the use of the short-time Fourier transform (STFT) method. The mu and beta frequency band spectral contents are made apparent by preserving the patterns of activation at various locations, times, and frequencies.

In our approach, the input data is used by the CNN [22] to learn the activation patterns of different MI signals. Convolution is applied only to the time axis, rather than frequency and location. Therefore, the shape of the activation patterns (i.e., power values from different frequencies) and their location (i.e., EEG channel) are learned in the convolutional layer. Then, a variational autoencoder (VAE) [23,24] with five hidden layers improves the classification through a deep network. Based on our current knowledge, EEG signals are characterized by a Gaussian distribution. For this reason, we used a decoder with Gaussian outputs to fit the distribution of EEG signals [25,26]. Learning an undercomplete representation forces the autoencoder to capture the most salient features of the EEG data. To compare the performance of the proposed CNN-VAE network, CNN-SAE, CNN, and VAE networks were also tested separately to assess their classification abilities.

BCI Competition IV dataset 2b was used in the analysis and evaluation of the methods proposed in the present study [27]. The results of the approaches described in this paper were compared with the results of current state-of-the-art algorithms used in this area of study. To test the proposed methods on another dataset, we also used these methods with the same testing protocol with our own dataset and compared the results with those of current state-of-the-art studies.

The rest of this paper is organized as follows. The dataset is presented in Section 2. Input data forms and deep learning networks, including the CNN, VAE, and combined CNN-VAE, are explained in Section 3. The experiment and its results are presented and discussed in Section 4. Finally, Section 5 concludes this study.

## 2. Datasets

There were two datasets of the two-class MI paradigm used in our experiments. The first dataset used in this experiment is BCI Competition IV dataset 2b [27]. Nine subjects participated in an experiment on MI by performing two different tasks (left hand, right hand). The EEG signals were recorded with three channels (C3, Cz, and C4), sampled at 250 Hz, and band-pass filtered between 0.5 and 100 Hz. The dataset includes three sessions in the training set. In the first two sessions, each trial started with the subject fixating on a cross and an additional short acoustic warning tone. Then, when t = 3 s, the MI task was indicated by an arrow, which was shown for 1.25 s. Afterward, the experimental subject was given 4 s for imagining the MI tasks (Figure 1a).

The second dataset includes data from five male subjects (ages from 23 to 30). During the experiment, these subjects were seated in front of a computer screen and followed instructions to perform two MI tasks (left hand, right hand). The EEG data used in the study were sampled at 250 Hz. The EEG used for processing was recorded with Ag-AgCl electrodes that were placed according to the extended international 10–20 system. The five channels used were C3, C1, Cz, C2, and C4. This dataset includes three sessions in the training set. Each trial was structured as follows (Figure 1b). First, a “rest” icon was shown on the screen to remind subjects to rest for 3 s with their arm relaxed. This is followed by a “prepare” indicator to provide subjects with 3 s to prepare for the task. Then, a “begin” icon was shown on screen for 0.5 s. Finally, one of two patterns was displayed for 6 s while the subject was performing MI tasks. The properties of both datasets are summarized in Table 1.

## 3. Deep Learning Framework

### 3.1. Input Form

The datasets that we used in this study include recordings from three electrodes (C3, Cz, and C4) and five electrodes (C3 C1, Cz, C2, and C4) to capture signals during left/right-handed MI task. These electrode points are located on the motor area of the brain.

It is shown in [5,21] that the power spectrum in mu band (6∼13 Hz) observed in the primary motor cortex of the brain decreases when performing an MI task. This desynchronization is called ERD. An MI task also causes the power spectrum in the beta band (13∼30 Hz) to increase, which is called ERS. The imagining of body movements is accompanied by a circumscribed event-related synchronization/desynchronization (ERS/ERD). Left-and right-handed movement can affect the signals in the right and left sides of the motor cortex at the C4 (C4, C2) and C3 (C3, C1) electrode sites, respectively. Cz is also affected by the MI task. With these facts, we designed a deep learning framework input that takes advantage of the time and frequency information of the data.

We extracted EEG signals with a length of 2 s from each MI EEG recording. Because the EEG signals were sampled at 250 Hz, each 2 s long time series corresponds to 500 samples. Then, the STFT was applied to the time series. The STFT was conducted with time lapses = 14 and window size = 64. Among all 500 samples, the STFT was computed for 32 windows on the first 498 samples, with the remaining 2 samples simply overlooked in the end. Therefore, a 257×32 image is produced, where the numbers 32 and 257 represent the samples on the axes of time and frequency, respectively. Subsequently, the beta and mu frequency bands were extracted from the spectrum of the output. Frequency bands of 6∼13 and 17∼30 were taken as the mu and beta bands, respectively. Despite the slight difference between these frequency bands and those presented in the literature, better data representation was obtained in the present experiment. The extracted images for beta and mu have sizes of 16×32 and 23×32, respectively. The cubic interpolation method was then used to resize the beta frequency band to 15×32 to ensure similar effects of both bands. After that, all the images were appropriately combined with each other and eventually made into an $N_{fr}\times N_{t}$ image, in which $N_{fr}=31$ and $N_{t}=32$.

There were $N_{c}=3$ electrodes (C3, Cz, and C4) used to measure the signals for BCI Competition IV dataset 2b, and the result was combined with the preserved neighboring information of the electrodes. As a result, an input image was obtained with a size of $N_{h}\times N_{t}=93$, in which $N_{h}=N_{fr}\times N_{c}=93$. In addition, in our own dataset, there were $N_{c}=5$ electrodes including C3, C1, Cz C2, and C4. So $N_{h}$ in our dataset equals $N_{fr}\times N_{c}=155$. Figure 2 demonstrates a sample input image which is composed of the MI task signals from three electrodes (a) and five electrodes (b). With the use of the proposed approach, brain activation on both sides of the brain motor cortex led to different patterns of activation in the brain vertical cortex.

### 3.2. Convolutional Neural Network (CNN)

A CNN is a neural network with multiple layers, including several convolution pooling layer pairs, as well as a completely connected output layer [22]. The design of the standard CNN aims at the recognition the image shapes can remain partly invariant to the shape location. In the convolutional layer, the input images were convolved with several 2-D filters. For example, if we use a 2-D image I as our input, we probably also want to use a 2-D convolutional kernel *K*:(1)$$S\left(i,j\right)=\left(I*K\right)\left(i,j\right)=\sum_{m}\sum_{n}K\left(i-m,j-n\right)$$

Then the feature map after convolutional kernel were down-sized into a smaller sample in the pooling layer. The network weights and filters(kernel) in the convolutional layer are learned by a back-propagation algorithm to reduce classification errors.

In our data, electrode location, time, and frequency data are all used at the same time. For the input images, activation at vertical locations is of great importance for the classification performance, while, in contrast, the horizontal locations of activation are not as significant. Hence, in the present experiment, the kernel applied were as high as the input image, while, on the horizontal axis, the filters applied were 1D throughout.

The network was used to train a total of $N_{F}=30$ filters of the size $N_{h}\times 3$. The proposed CNN structure is shown in Figure 3. The input images were convolved with the kernels to be trained and were then put through *f*-the output function-to generate the map of output in the convolution layer. In a given layer, the map of *k*-th features can be obtained as
(2)$$h_{ij}^{k}=f\left(a\right)=f\left({\left(W^{k}*x\right)}_{ij}+b_{k}\right)$$
where x denotes the input image, $W^{k}$ represents the weight matrix for filter *k* and $b_{k}$ represents the bias value, for $k=1,2,\cdots ,N_{F}.$ In the present experiment where the 1D filtering has $N_{h}\times 3$ sized filters, j=1 and $i=1,2,\cdots ,N_{t}-2$. The output function *f* is selected as rectified linear unit (ReLU) function. ReLU is defined as
(3)$$f\left(a\right)=\mathrm{ReLU}\left(a\right)=\ln\left(1+e^{a}\right)$$
where a is as defined in Equation (Equation 1).

The output of the convolutional layer is $N_{F}$ vectors with $(N_{t}-2)\times 1$ dimension. In the max-pooling layer, zero padding and a sampling factor of 10 are applied. Hence, the map of the output obtained in the previous layer is subsampled into $N_{F}$ vectors with 1D. The layer following the max-pooling layer is a completely connected layer with two outputs that represent the MI of the right and left hand, respectively. The back-propagation algorithm is used to learn CNN parameters. With the proposed approach, the network is fed with the labeled training set, and the error *E* is computed taking into account the fact that the desired output is different from the output of the network. Subsequently, the gradient descent method is applied to minimize the error E occurring with changes in the network parameters, which is demonstrated in Equations (4) and (5).
(4)$$W^{k}=W^{k}-\eta \frac{tialE}{tialW^{k}}$$
(5)$$b_{k}=b_{k}-\eta \frac{tialE}{tialb_{k}}$$
where η is the learning rate of the algorithm, while $W^{k}$ represents the weight matrix for kernel *k* and $b_{k}$ represents the bias value, just like the previous definition. At last, the network that has been trained is used to classify the new samples in the test set.

### 3.3. Variational Autoencoder (VAE)

An autoencoder (AE) is a neural network that is trained to attempt to copy its input to its output. Internally, it has a hidden layer h that describes a code used to represent the input. The network may be viewed as consisting of two parts: an encoder function z=f(x) and a decoder that produces a reconstruction r=g(z). Where *x* is the input data. One way to obtain useful features from the autoencoder is to constrain *z* to have smaller dimension than *x*. An autoencoder whose code dimension is less than the input dimension is called undercomplete. Learning an undercomplete representation forces the autoencoder to capture the most salient features of the training data.

The variational autoencoder (VAE) is a directed model that uses learned approximate inference and can be trained purely with gradient-based methods [23].

To generate a sample from the model, the VAE first draws a sample *z* from the code distribution $p_{model}\left(z\right)$. The sample is then run through a differentiable generator network g(z). Finally, *x* is sampled from a distribution $p_{model}\left(x;g\left(z\right)\right)=p_{model}\left(x|z\right)$. However, during training, the approximate inference network (or encoder) q(z|x) is used to obtain *z* and $p_{model}\left(x|z\right)$ is then viewed as a decoder network. The key insight behind variational autoencoders is that they may be trained by maximizing the variational lower bound L(q) associated with data point *x*:(6)$$\begin{array}{cc} L(q)= & E_{z∼q(z|x)}\log p_{model}\left(z,x\right)+H\left(q\left(z|x\right)\right) \end{array}$$
(7)$$\begin{array}{cc} = & E_{z∼q(z|x)}\log p_{model}\left(x|z\right)-D_{KL}\left(q\left(z|x\right)|\right|p_{model}\left(z\right)) \end{array}$$

In Equation (Equation 6), we recognize the first term as the joint log-likelihood of the visible and hidden variables under the approximate posterior over the latent variables. We recognize also a second term, the entropy of the approximate posterior. When *q* is chosen to be a Gaussian distribution, with noise added to a predicted mean value, maximizing this entropy term encourages increasing the standard deviation of this noise. More generally, this entropy term encourages the variational posterior to place high probability mass on many *z* values that could have generated *x*, rather than collapsing to a single point estimate of the most likely value. In Equation (Equation 7), we recognize the first term as the reconstruction log-likelihood found in other autoencoders. The second term tries to make the approximate posterior distribution q(z|x) and the model prior $p_{model}\left(z\right)$ approach each other. The network described in Equation (Equation 7) is much like the network shown in Figure 4 (left). Stochastic gradient descent on back-propagation can handle stochastic inputs, but not stochastic units within the network. The solution, called the “reparameterization trick”, is to move the sampling to an input layer. we can sample from N(μ(x),θ(x)) by sampling ϵ∼N(0,I), then computing $p_{model}z=\mu \left(x\right)+\theta^{1/2}\left(x\right)*\epsilon$. Where μ(x) and θ(x) are the mean and covariance of q(z|x). Thus, Equation (Equation 7) can be computed as
(8)$$\begin{array}{cc} L(q)= & E_{\epsilon ∼N(0,I)}p_{model}\left(x|z=\mu \left(x\right)+\theta^{1/2}\left(x\right)\times \epsilon \right)-D_{KL}\left(q\left(z|x\right)|\right|p_{model}\left(z\right)) \end{array}$$

This is shown schematically in Figure 4 (right) VAE is composed of an input layer, several AEs, and an output layer. Each AE layer is trained separately in an unsupervised manner, and the output of the hidden layer in the previous AE is used as input to the next layer in the deep network. After this unsupervised pre-training step, a supervised fine-tuning step is applied to learn the whole network’s parameters by using the back-propagation algorithm [22]. The VAE network used in this study is shown in the VAE part of Figure 4. This model is composed of one input layer, five hidden layers, and one output layer. The number of nodes at each layer is indicated at the top of the figure.

### 3.4. Combined CNN-VAE

The useful information from an EEG signal is very weak and thus susceptible to noise [7]. Artifacts, such as eye-blinking and muscle movement, are another source of disturbance that cause irrelevant effects that corrupt the desired brain pattern. The channel correlation, presence of artifacts (i.e., movement), and high dimensionality of EEG data make it challenging to design an ideal EEG classification framework. To address these issues, we propose a new deep learning model that implements a CNN followed by a VAE. The model is presented in Figure 5.

In this deep learning model, at first the CNN Introduced in Section 3.2 is employed over the input data and the kernels and network parameters are learned. Then, the CNN output is used as an input to the VAE network. Input layer of the VAE has 900 neurons which is the output of 30 neurons in the convolutional layer for each of 30 kernels trained in CNN. By using this framework, we aim to improve the classification accuracy by using CNN-VAE model to consider the time, frequency, and position information of EEG data.

The computational complexity of CNN-VAE used for testing is O($N_{h}\times N_{t}\times N_{F}\times N_{Fs}$) + O (${\left(N_{t}\times N_{F}\right)}^{2}\times N_{L}$). The first term is the computational complexity of the CNN portion of the network, where $N_{h}\times N_{t}$ is the input image size, $N_{Fs}$ is the number of convolution kernels, and $N_{F}$ is the kernel size. The second term is the computational complexity of the VAE part, where $N_{L}$ is the number of layers; since the VAE layer is fully connected, the complexity has a squared term.

## 4. Results and Discussion

In this section, the results of the proposed approaches on these datasets are presented.

### 4.1. Results

In BCI Competition IV dataset 2b, separate training and testing sets were given to each classifier for all experimental subjects. The proposed approaches’ performance and mean kappa value metrics were accurately evaluated using 10×10 fold cross-validation. Among the 400 trials in every session, 90% were selected for the training set in a random manner, with the remaining 10% assigned to the test set. This series of operations was repeated 10 times. Then, for all EEG trials, a time interval of 0.5–2.5 s was extracted after the display of the cue. Subsequently, the extracted signals were processed with the application of the STFT, and the input was constructed according to the description in Section 3.1.

In the CNN approach, a network with one max-pooling layer and convolutional layer was used, as demonstrated in Figure 3 and Section 3.2. A batch training method was used to train this network, using a batch size of 50 for 300 epochs. For the VAE method, the input image, which is presented in Figure 1, was down-sampled and transformed into a 900 × 1 vector, which allows it to be used as the VAE network input. As described in Section 3.3, for the classification, a network with five hidden layers was applied. All the VAE autoencoders were trained for 200 epochs with a batch size of 20. Subsequently, fine-tuning was employed for 200 epochs with a batch size of 40. For the computation of the values of the convolutional layer of the CNN, filters and other parameters of the network were applied. The obtained values were subsequently turned into vectors following the description in Section 3.4, and the vectors were then used as the VAE network input.

The kappa value is a popular measure for EEG classification performance which can remove the effect of accuracy of random classification. Kappa is calculated as
(9)$$\mathrm{kappa}=\frac{\mathrm{acc}-\mathrm{rand}}{1-\mathrm{rand}}$$

The kappa results of the FBCSP, CNN, CNN-SAE, and CNN-VAE methods are presented in Table 2; FBCSP is a competition-winning algorithm [10]. Table 2 also presents the standard deviation of kappa values for 10 sessions in the 10×10 cross-validation process. As is clear from Table 2, the average kappa result for the CNN-VAE method is 0.564 is higher than that for the other methods. For five of the nine subjects, the kappa values of the CNN-VAE method are the best. Furthermore, the standard deviation is 0.065 for the CNN-VAE method. Even though FBCSP presents a smaller standard deviation of 0.014, the standard deviation of the CNN-VAE method is still small enough to be considered robust during EEG signal-processing.

To evaluate our approach using another dataset, we used the same network explained earlier to classify our own datasets. To evaluate our method, we also evaluated the effect of the number of channels on the results. Our dataset is divided into two datasets: three electrodes (C3, Cz, C4) and five electrodes (C3, C1, Cz, C2, C4). In both datasets, we extracted the 1–3 s time interval from the motor imagination task data. The input images were constructed as explained in Section 3.1, with a window size of 32 and time lapses of 7. Furthermore, among the 400 trials in every session, 90% were selected for the training set in a random manner, with the remaining 10% composing the test set.

The results for our own dataset with three electrodes (C3, Cz, C4) and five electrodes (C3, C1, Cz, C2, C4) are shown in Table 3 and Table 4, respectively. The average kappa results of the CNN-VAE method with three electrodes is 0.568, which is higher than that of the other methods, and the corresponding standard deviation is 0.068. Furthermore, in the five-electrode situation, the average kappa result of the CNN-VAE method is 0.603, which is higher than that of the other methods, and the corresponding standard deviation is 0.067. We applied Kolmogorov-Smirnov test to determine whether the results follow a normal distribution. As clearly present, the kappa value of 5 electrodes is better than the value of 3 electrodes. However, the standard deviations of both different number of electrodes situation are almost the same. We applied Kolmogorov-Smirnov test to determine whether the results follow a normal distribution. As shown in Table 5, all the *p*-value is greater than 0.05. Furthermore, to evaluate CNN-VAE approach among the compared methods, ANOVA is applied. Table 6 shows the *p*-value between each two methods. Figure 6a–c show the ANOVA stats of different methods on BCI Competition IV dataset 2b, our own dataset with 3 electrodes and 5 electrodes, respectively. There is no statistically significant differences main effect of classification methods.

Based on the results, CNN-VAE method has higher level of robustness to the subject dependent differences, and the CNN-VAE framework an effective and good classification method. In addition, increased number of channels has a positive impact on classification performance.

### 4.2. Discussion

The most distinctive part of this method is the use of VAE. VAE uses a variational method to fit a Gaussian distribution to encode the hidden layer. This requires the EEG signal to conform to the Gaussian distribution. In this study, the Kolmogorov-Smirnov test was used to analyze the EEG signals. In Figure 7a is the average distribution of EEG signals of all subjects on BCI Competition IV dataset 2b, and Figure 7b is the average distribution of EEG signals of all subjects on our own dataset. The Kolmogorov-Smirnov test shows that both BCI Competition IV dataset 2b (p=0.318>0.05) and on our own dataset (p=0.537>0.05) follow a Gaussian distribution.

Different subjects exhibit different sensitivities when performing MI tasks. This results in a large difference among classification performances for different subjects. For BCI Competition IV dataset 2b, the input images of two classes are shown Figure 8 for subject 2 and subject 4. The difference between left-handed and right-handed MI tasks is quite clear from these average images for subject 4. While there is an increase in the C3 area and decrease in the C4 area for the left-handed MI, this is the opposite for the right-handed MI. Nevertheless, average images of the MI test are not that different for subject 2. Correspondingly, using the CNN-VAE method, the kappa value for subject 4 is the best at 0.908, and the value for subject 2 is the worst at 0.346. The kappa values for subject 4 and subject 2 are 0.888 and 0.208, respectively, when using FBCSP. Therefore, the improvement for subject 2 is much more than the improvement for subject 4 when using CNN-VAE. This shows that CNN-VAE has better applicability.

Using our own dataset, the best performance is for subject Z, with kappa values of 0.768 and 0.821 with three and five electrodes, respectively. The input images of these two subjects are shown in Figure 9. Obviously, the images for subject Z are much better than the ones for subject D.

The effect of epoch size on performance (kappa) is shown in Figure 10 along with the average training time (for one training set with 360 trials). As can be seen in the figure, for the appropriate calculation time (226 s), the performance with 100 epochs is higher than that with other period lengths. The number of epochs selected in this study was 100. The size of the kernel in the convolutional layer also changes the performance of the classification. A comparison between the different kernel size performances is given in Table 6. Since the performance of the kernel is significantly reduced by increasing its size, the results of the larger kernel were not studied. Please note that Nh is fixed due to 1D filtering, and its value is the height of the input image. As shown in Table 7, the convolution layer with a kernel size equal to $N_{h}\times 3$ achieves the best classification performance. In our experiments, the CNN-VAE network of the $N_{h}\times 3$ kernel has higher performance. This means that using this kernel size can better represent these features.

The effect of the number of channels on the results is an issue to be addressed in our dataset. As shown in Figure 11, the result with the five channels is significantly better than that with the three channels, yet the effect of the number of channels on the standard deviation is not substantial. In future work, we will delve into the effects of channel number and channel position on performance.

In this study, the CNN-VAE model takes advantage of the multidimensional (time, frequency, and channel) features of the CNN model while gaining the benefits of the VAE method. This imparts the CNN-VAE framework with a great performance in MI classification.

## 5. Conclusions

In this paper, we propose a new approach that combines the CNN and VAE networks. In this framework, the input images built by EEG signals are first trained for feature extraction by the CNN. After that, the extracted features are classified by the VAE model. After comparing the results of the CNN-VAE method with those of other existing approaches, it is shown that the approach in our study yields the most accurate results using BCI Competition IV dataset 2b and our own dataset. Yet there are no statistically significant differences among the compared methods. In future work, some new approaches will be employed to overcome the low signal-to-noise ratio of EEG signals.

The filters of the CNN consider neighboring regions, yet the CNN provides no such information, which suggests a greater contribution of the filters to the performance of classification compared with their role in the other methods. Furthermore, the undercomplete representation by the VAE follows the Gaussian distribution, which is the distribution of EEG signals. Therefore, when the outputs of the CNN model are input into the VAE, the advantages of the two networks are combined, thus increasing the ultimate classification performance. In conclusion, CNN-VAE is a novel, promising framework for the classification of EEG signals generated during MI tasks.

## Figures and Tables

**Figure 1 sensors-19-00551-f001:**
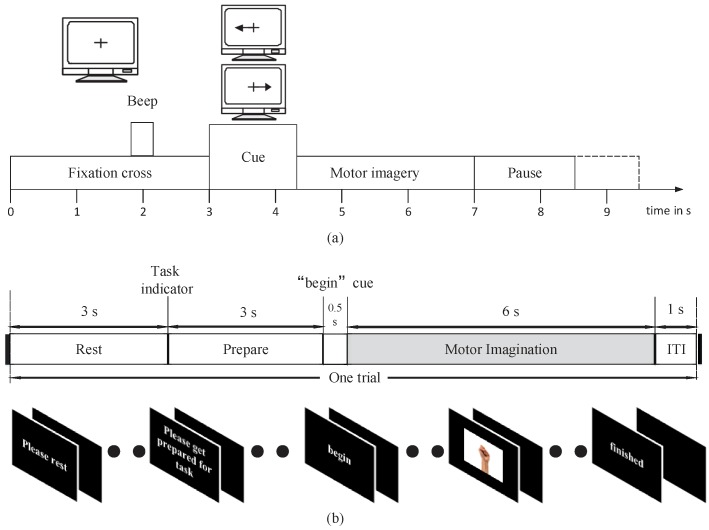
Diagram of a trial and timings during a session of BCI Competition IV dataset 2b (**a**) and our own dataset (**b**).

**Figure 2 sensors-19-00551-f002:**
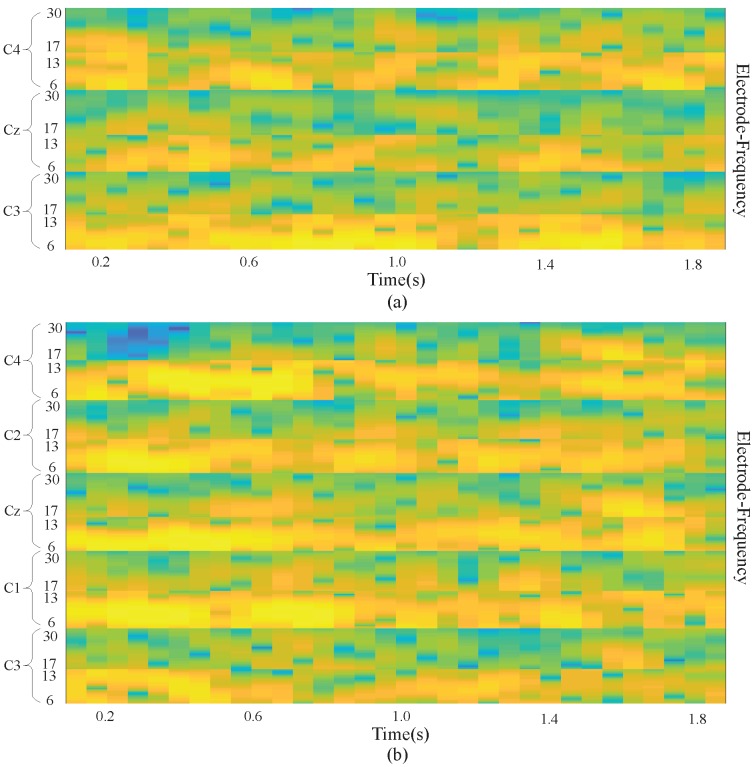
Input image with 3 electrodes and 5 electrodes. (**a**) is the input image with 3 electrodes; (**b**) is the input image with 5 electrodes

**Figure 3 sensors-19-00551-f003:**
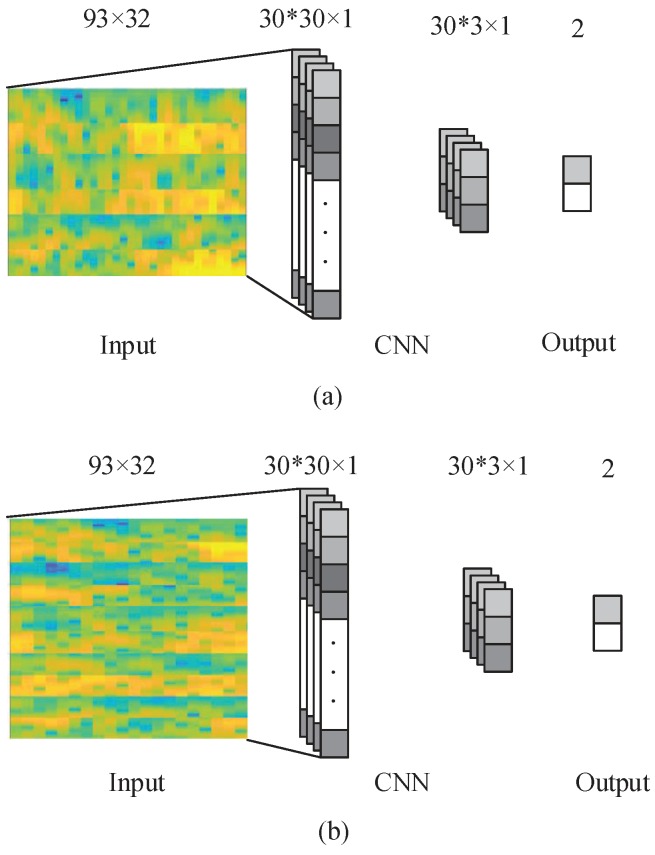
The proposed CNN network with 3 electrodes and 5 electrodes. (**a**) is the CNN network with 3 electrodes; (**b**) is the CNN network with 5 electrodes.

**Figure 4 sensors-19-00551-f004:**
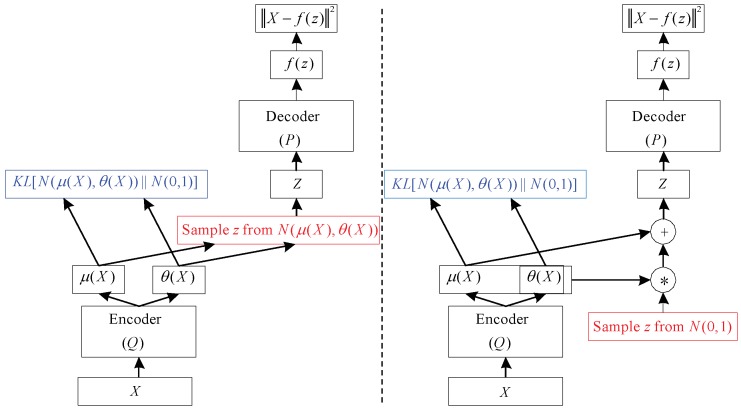
A training-time variational autoencoder implemented as a feedforward neural network. (**Left**) is without the “reparameterization trick”, and (**right**) is with it [24].

**Figure 5 sensors-19-00551-f005:**
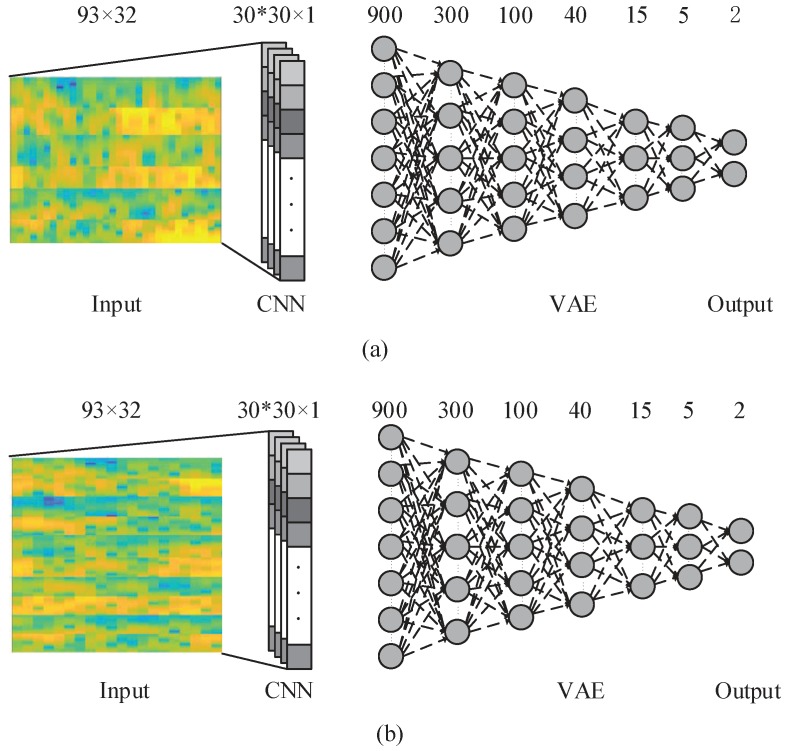
The proposed CNN-VAE network with 3 electrodes and 5 electrodes. (**a**) is the CNN-VAE network with 3 electrodes; (**b**) is the CNN-VAE network with 5 electrodes.

**Figure 6 sensors-19-00551-f006:**
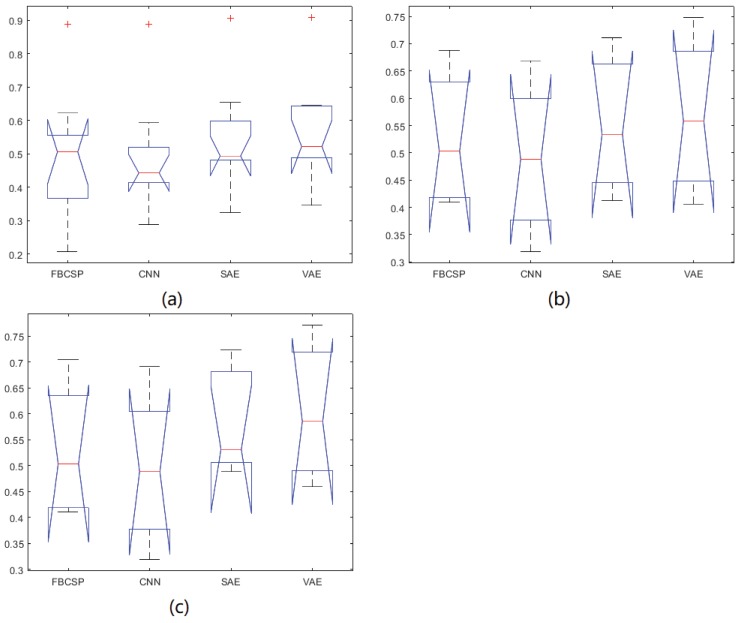
The ANOVA stats of different methods.(**a**) is The ANOVA stats of different methods on BCI competition IV dataset 2b; (**b**) is The ANOVA stats of different methods on our own dataset with 3 electrodes; (**c**) is The ANOVA stats of different methods on our own dataset with 5 electrodes.

**Figure 7 sensors-19-00551-f007:**
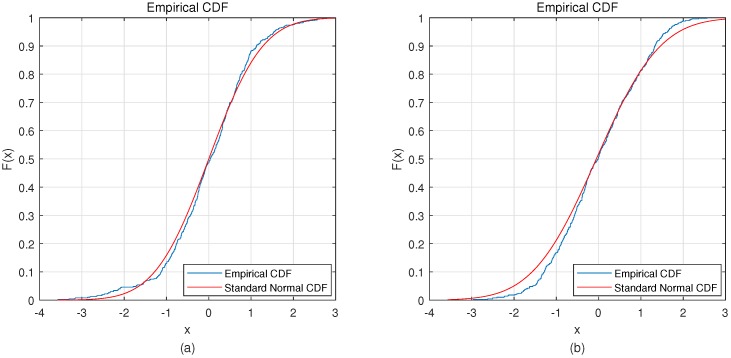
Distribution of both datasets. (**a**) is the distribution of BCI competition IV dataset 2b; (**b**) is the distribution of our own dataset.

**Figure 8 sensors-19-00551-f008:**
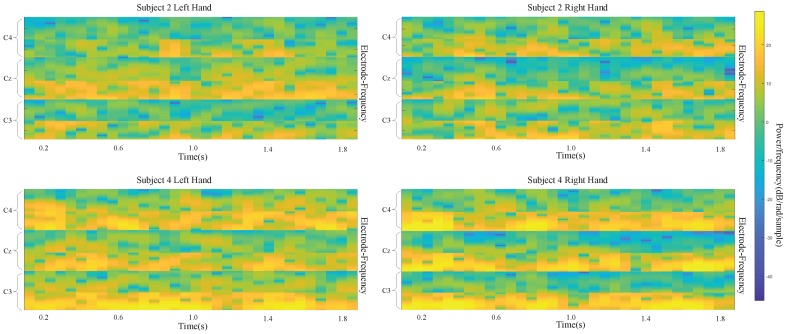
input images for left- and right-hand classes of subject 2 and subject 4.

**Figure 9 sensors-19-00551-f009:**
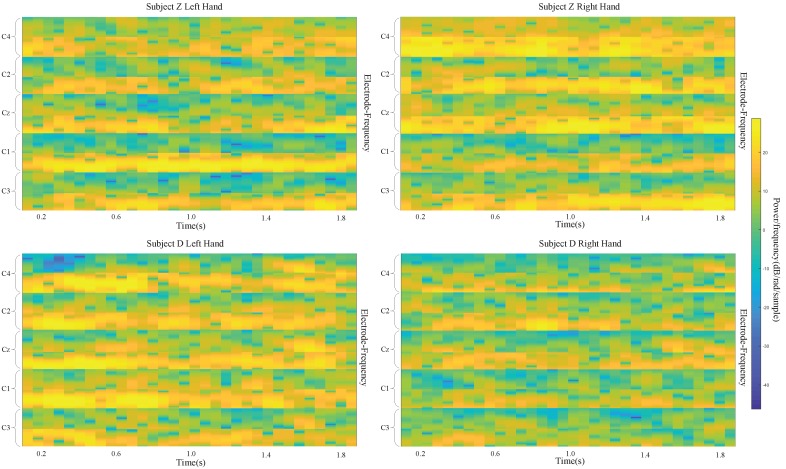
input images for left- and right-hand classes of subject D and subject Z.

**Figure 10 sensors-19-00551-f010:**
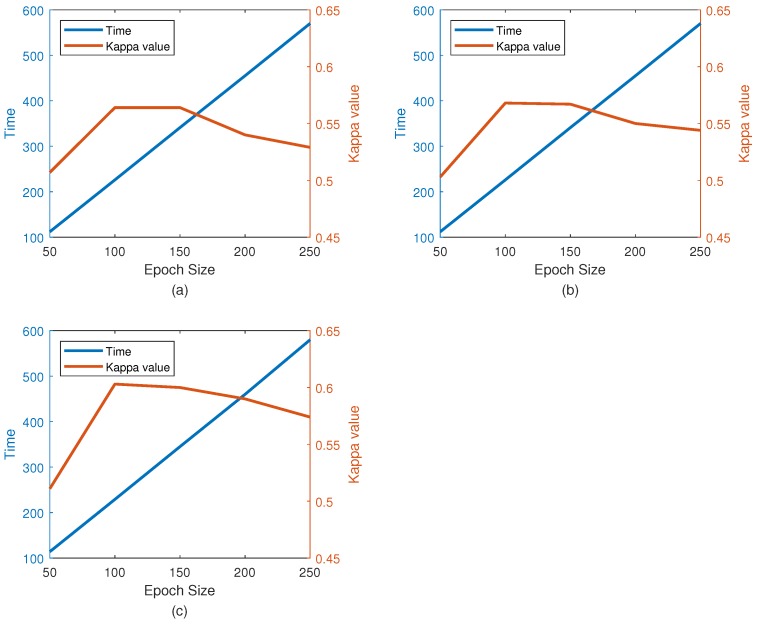
Effect of number of epoch on kappa value and training time. (**a**) The performance on BCI Competition IV dataset 2b; (**b**) The performance on our own dataset with 3 electrodes; (**c**) The performance on our own dataset with 5 electrodes.

**Figure 11 sensors-19-00551-f011:**
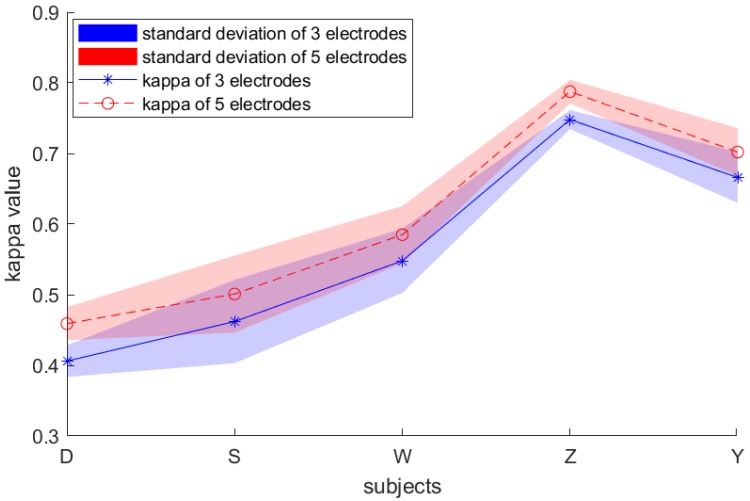
Kappa value and standard deviation results of our own dataset with 3 electrodes and 5 electrodes.

**Table 1 sensors-19-00551-t001:** Data description for both datasets.

Dataset	Subjects	Channels	Trials
Competition IV dataset 2b	9	C3, Cz, C4	400
Our own dataset	5	C3, C1, Cz, C2, C4	400

**Table 2 sensors-19-00551-t002:** Comparison of kappa value and standard deviation for CNN-VAE and 3 Competitive methods on BCI competition IV dataset 2b. (The data in bold is the best performance among the different methods).

Subject	BCI Competition III (Mean ± Std. Dev.)			
	FBCSP	CNN	CNN-SAE	CNN-VAE
1	**0.546 ± 0.017**	0.488 ± 0.158	0.517 ± 0.095	0.522 ± 0.076
2	0.208 ± 0.028	0.289 ± 0.068	0.324 ± 0.065	**0.346 ± 0.068**
3	0.244 ± 0.023	0.427 ± 0.071	**0.494 ± 0.084**	0.436 ± 0.060
4	0.888 ± 0.003	0.888 ± 0.008	0.905 ± 0.017	**0.908 ± 0.009**
5	0.692 ± 0.005	0.593 ± 0.083	**0.655 ± 0.060**	0.646 ± 0.075
6	0.534 ± 0.012	0.495 ± 0.073	0.579 ± 0.099	**0.642 ± 0.057**
7	0.409 ± 0.013	0.409 ± 0.079	0.488 ± 0.065	**0.550 ± 0.072**
8	0.413 ± 0.013	0.443 ± 0.133	0.494 ± 0.106	**0.506 ± 0.083**
9	**0.583 ± 0.010**	0.415 ± 0.050	0.463 ± 0.152	0.518 ± 0.078
Average	0.502 ± 0.014	0.494 ± 0.080	0.547 ± 0.083	**0.564 ± 0.065**

**Table 3 sensors-19-00551-t003:** Comparison of kappa value and standard deviation for CNN-VAE and 3 Competitive methods on our own dataset with 3 electrodes. (The data in bold is the best performance among the different methods).

Subject	Kappa Value (Mean ± Std. Dev.)			
	FBCSP	CNN	CNN-SAE	CNN-VAE
D	0.410 ± 0.029	0.319 ± 0.057	**0.412 ± 0.038**	0.406 ± 0.045
S	0.421 ± 0.061	0.396 ± 0.149	0.456 ± 0.132	**0.462 ± 0.118**
W	0.503 ± 0.022	0.488 ± 0.097	0.534 ± 0.084	**0.558 ± 0.081**
Z	0.688 ± 0.008	0.669 ± 0.035	0.711 ± 0.036	**0.748 ± 0.027**
Y	0.611 ± 0.037	0.576 ± 0.074	0.647 ± 0.068	**0.666 ± 0.072**
Average	0.527 ± 0.031	0.490 ± 0.082	0.552 ± 0.072	**0.568 ± 0.068**

**Table 4 sensors-19-00551-t004:** Comparison of kappa value and standard deviation for CNN-VAE and 3 Competitive Methods our own dataset with 5 electrodes. (The data in bold is the best performance among the different methods).

Subject	Kappa Value (Mean ± Std. Dev.)			
	FBCSP	CNN	CNN-SAE	CNN-VAE
D	0.410 ± 0.022	0.319 ± 0.065	**0.488 ± 0.040**	0.459 ± 0.047
S	0.421 ± 0.056	0.396 ± 0.138	**0.512 ± 0.122**	0.501 ± 0.109
W	0.503 ± 0.029	0.488 ± 0.107	0.531 ± 0.084	**0.585 ± 0.081**
Z	0.704 ± 0.006	0.691 ± 0.050	0.723 ± 0.046	**0.771 ± 0.031**
Y	0.611 ± 0.030	0.576 ± 0.085	0.668 ± 0.075	**0.702 ± 0.068**
Average	0.530 ± 0.028	0.494 ± 0.089	0.584 ± 0.073	**0.603 ± 0.067**

**Table 5 sensors-19-00551-t005:** The results difference to a normal distribution.

Dataset	FBCSP (*p*-Value)	CNN (*p*-Value)	CNN-SAE (*p*-Value)	CNN-VAE (*p*-Value)
BCI Competition IV dataset 2b	0.116	0.653	0.534	0.434
Our own dataset with 3 electrodes	0.188	0.619	0.479	0.434
Our own dataset with 5 electrodes	0.174	0.598	0.479	0.416

**Table 6 sensors-19-00551-t006:** The results difference to a normal distribution.

Dataset	CNN-VAE vs. CNN-SAE (*p*-Value)	CNN-VAE vs. CNN (*p*-Value)	CNN-VAE vs. FBCSP (*p*-Value)	CNN-SAE vs. CNN (*p*-Value)	CNN-SAE vs. FBCSP (*p*-Value)	CNN vs. FBCSP (*p*-Value)
BCI Competition IV dataset 2b	0.822	0.381	0.362	0.510	0.473	0.901
Our own dataset with 3 electrodes	0.854	0.402	0.632	0.479	0.753	0.666
Our own dataset with 5 electrodes	0.805	0.249	0.392	0.293	0.428	0.690

**Table 7 sensors-19-00551-t007:** Effect of kernel size on CNN. $N_{h}$ = 93, 93, and 155. (The data in bold is the best performance among the different methods).

Dataset	Kernel Size	Kappa	Time (s)
BCI Competition IV dataset 2b	$N_{h}\times 1$	0.492	154
	$N_{h}\times 2$	0.546	191
	$N_{h}\times 3$	**0.564**	226
	$N_{h}\times 4$	0.521	274
	$N_{h}\times 5$	0.521	325
Our own dataset with 3 electrodes	$N_{h}\times 1$	0.500	153
	$N_{h}\times 2$	0.543	190
	$N_{h}\times 3$	**0.568**	226
	$N_{h}\times 4$	0.526	274
	$N_{h}\times 5$	0.522	325
Our own dataset with 5 electrodes	$N_{h}\times 1$	0.525	155
	$N_{h}\times 2$	0.572	190
	$N_{h}\times 3$	**0.603**	227
	$N_{h}\times 4$	0.562	275
	$N_{h}\times 5$	0.550	326

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
