# Peer review of "EEG Classification of Motor Imagery Using a Novel Deep Learning Framework"

_sensors, 2019, doi:10.3390/s19030551_

Round 1

Reviewer 1 Report

Using artificial neural networks to learn EEG data for BCI is an interesting topic, and the authors proposed a combination of CNN and autoencoder networks, and showed the overall improvement in detection accuracy over FBCSP, CNN, and CNN-SAE.

I have a few comments as follows.

1) Understand that CNN-SAE ([21]) is a closely-related prior art. How does CNN-VAE differ from CNN-SAE, in theory and in practice? From the classification result (Tables 2~4), CNN-SAE seems to just marginally lose to CNN-VAE in terms of Kappa value. Is that margin significant? 

Please also show the comparison in terms of accuracy rate.

2) The manuscript needs to be substantially improved in terms of technical writing. Given there are many relevant prior arts, there is however, no identification of technical gap that drove this research.

3) Manuscript text and graphs were not well prepared: as in either Figure 5 or Figure 3, both upper and lower panels show the same pictures except the input images are different (whilst the given numbers are identical). Some graphs such as Figure 4 lacks explanation: what are the left and the right networks?

3) Many statements are not accurate, nor specific, sometimes quite arbitrary in language use. 

 ------"STFT can extract frequency information from channel information to make full use of electrodes resources." (abstract). 

----- "to fit the distribution of EEG signal which is Gaussian distributed too." (abstract)  why is the signal (VAE input) Gaussian?

Author Response

Thank you for your comments. Attached please find the response letter.

Reviewer 2 Report

A classifier based on the combination of short time Fourier transform (STFT), a convolutional neural network (CNN) architecture and  a Variationa autoencoder (VAE) is proposed for EEG classification in motor imagery (MI) brain computer interfaces (BCI). The paper provides an introduction including references to some of the relevant papers in the topics involved in the paper, and evaluates the achieved accuracy by using two different datasets (one is well known in the literature and the other has been built by the authors although is available from the authors) and comparing the accuracy (by using the Kappa index) with other approaches previously described in the literature.

Nevertheless, to be publishable, some improvements should be included in the paper:

- The list of references should be revised to include more recent papers and the introduction should be modified according to it. For example, with respect to reference number 15, there is a new review of classification algorithms for EEG-based BCI:

Lotte F, Bougrain L, Cichocki A, Clerc M, Congedo M, Rakotomamonjy A, Yger F. A review of classification algorithms for EEG-based brain-computer interfaces: a 10 year update. J Neural Eng. 2018 Jun;15(3):031005. doi: 10.1088/1741-2552/aab2f2. 2018 Feb 28.

-The performance of the procedure is compared with other approaches by using the accuracy (through the kappa index). Nevertheless, it is also important to compare the complexity of the approaches, for example, by using measures of the time required for training the classifiers. The conclusions of the paper should be completed with the information obtained from these results.

- The mathematical descriptions of the CNN and VAE and the corresponding learning algorithms should be more detailed.

- The hyperparameters of the CNN and VAE seem to be selected ad hoc. Some explanation about how they can be obtained (to make the procedure applicable to other datasets) and some analysis about the change in the accuracy observed after changing the values of some hyperparameters should be included in the paper.  

- Perhaps the main concern is related with the statistical significance of the observed differences between methods compared. Although the average and the standard deviation are considered in the paper, some test of statistical significance should be applied (for example, after determining whether the results follow a normal distribution by applying the Kolmogorov-Smirnov test, an ANOVA or a Kruskal-Wallis test could be applied to determine if the differences are statistically relevant. The conclusions of the paper should be completed with the information obtained from this analysis.

- A revision of the English is required to avoid grammar errors and typos.

Author Response

(The authors gave the same response as above.)

Reviewer 3 Report

In this paper, a classification method based on combining CNN with VAE using EEG signals is proposed. Regarding the evaluation of the obtained kappa value, there are many vague expressions such as “to a certain degree” and “relatively more reliable and accurate.” The authors should make quantitative assessments by checking if there are any statistically significant differences. Also, in Tables 2 to 4, the kappa values in the best value is in bold, but for Line 1 in Table 4, 0.459 in bold is not the highest value.

Author Response

(The authors gave the same response as above.)

Round 2

Reviewer 2 Report

My comments and concerns corresponding to the first version has been considered and correctly answered in all the cases but in the issues regarding the statistical analyais of results: once the Kolmogorov-Smirnof test has been applied to check whether the results follow a normal distribution or not (this has been done), an analysis of variance is required to check the statistical significance of the differences observed among the results of the compared alternatives. Once this is checked, the conclusions should be revised properly.

Minor quesrions:

NF in line 212 should be N_F as in the expression of line 210 and 211

Author Response

Thank you for your comments. I will make the following changes with the attachment, and highlight the reward part in the review list and manuscript.

Round 3

Reviewer 2 Report

The interest on doing the ANOVA analysis is to determime whether there are statistically significant differences among the compared methods. Thus, the p-values for all the pairs of alternatives compared should be provided. In case of p-values below 0.05 there are significant differences. This analysis has not been included in the paper yet. Indeed, Figure 6 seems to indicate that there are no differences among the compared alternatives (and this is not good as the new method does not suppose any improvement. Please, include the p-values and the corresponding conclusions for the pairs of compared alternatives.

The figure caption of Figure 6 should include the meaning of figures 6.a, 6.b and 6.c 

Author Response

Thank you for your comments.

I will make the following changes.
